# Microbial Virulence Factors, Antimicrobial Resistance Genes, Metabolites, and Synthetic Chemicals in Cabins of Commercial Aircraft

**DOI:** 10.3390/metabo13030343

**Published:** 2023-02-24

**Authors:** Xi Fu, Mei Zhang, Yiwen Yuan, Yang Chen, Zheyuan Ou, Zailina Hashim, Jamal Hisham Hashim, Xin Zhang, Zhuohui Zhao, Dan Norbäck, Yu Sun

**Affiliations:** 1Guangdong Provincial Engineering Research Center of Public Health Detection and Assessment, School of Public Health, Guangdong Pharmaceutical University, Guangzhou 510006, China; 2Guangdong Provincial Key Laboratory of Protein Function and Regulation in Agricultural Organisms, College of Life Sciences, South China Agricultural University, Guangzhou 510642, China; 3Guangdong Laboratory for Lingnan Modern Agriculture, South China Agricultural University, Guangzhou 510642, China; 4Department of Environmental and Occupational Health, Faculty of Medicine and Health Sciences, Universiti Putra Malaysia, UPM, Serdang 43400, Malaysia; 5Universiti Selangor, Shah Alam 40000, Malaysia; 6Institute of Environmental Science, Shanxi University, Taiyuan 030006, China; 7Key Laboratory of Public Health Safety of the Ministry of Education, NHC Key Laboratory of Health Technology Assessment, School of Public Health, Fudan University, Shanghai 200032, China; 8Shanghai Key Laboratory of Meteorology and Health, Typhoon Institute/CMA, Shanghai 200030, China; 9Occupational and Environmental Medicine, Department of Medical Science, University Hospital, Uppsala University, 75237 Uppsala, Sweden

**Keywords:** microbial metabolomics, metagenomics, aircraft cabin, public health, surface materials

## Abstract

Passengers are at a higher risk of respiratory infections and chronic diseases due to microbial exposure in airline cabins. However, the presence of virulence factors (VFs), antimicrobial resistance genes (ARGs), metabolites, and chemicals are yet to be studied. To address this gap, we collected dust samples from the cabins of two airlines, one with textile seats (TSC) and one with leather seats (LSC), and analyzed the exposure using shotgun metagenomics and LC/MS. Results showed that the abundances of 17 VFs and 11 risk chemicals were significantly higher in TSC than LSC (*p* < 0.01). The predominant VFs in TSC were related to adherence, biofilm formation, and immune modulation, mainly derived from facultative pathogens such as *Haemophilus parainfluenzae* and *Streptococcus pneumoniae*. The predominant risk chemicals in TSC included pesticides/herbicides (carbofuran, bromacil, and propazine) and detergents (triethanolamine, diethanolamine, and diethyl phthalate). The abundances of these VFs and detergents followed the trend of TSC > LSC > school classrooms (*p* < 0.01), potentially explaining the higher incidence of infectious and chronic inflammatory diseases in aircraft. The level of ARGs in aircraft was similar to that in school environments. This is the first multi-omic survey in commercial aircraft, highlighting that surface material choice is a potential intervention strategy for improving passenger health.

## 1. Introduction

Public transportation vehicles, including buses, trains, subways, and aircraft, are often crowded and enclosed environments. Among these, aircraft is the most important mode of long-distance mass transit, and it can quickly transport passengers to various locations. However, taking aircraft can increase the risk of infectious and chronic inflammatory diseases. For example, increased SARS and COVID-19 infection in aircraft have been well documented [1,2]. A survey of 4316 passengers reported adverse health effects after flights, including dry mouth, dry eyes, nasal stuffiness, and respiratory inflammation [3]. A health survey among Swedish commercial pilots reported that the incidence and prevalence of asthma and allergy among pilots were 140% and 46% higher than in the general population in Sweden [4]. Mucosal irritation was commonly reported among aircrew, including itching/burning eyes (44.6%), stuffy/runny nose (39.6%), and itching/red skin (17.1%) [5]. 

Microorganisms pose a significant risk to the health of passengers and aircrew in aircraft. Pathogens such as SARS-CoV-2 can spread easily in crowded and enclosed aircraft cabins. Microbial species associated with respiratory inflammation, including *Haemophilus* and *Streptococcus*, are also commonly present in aircraft cabins [6]. Microbial volatile organic compounds (MVOCs), produced through the metabolism of fungi and bacteria, are a source of exposure to many chronic diseases and symptoms, such as headaches, fatigue, eye and nose irritation, and asthma [7]. The concentration of MVOCs in aircraft cabins is four times higher than in the average home environment [8]. Pentanols, an MVOC that causes skin and eye irritation, are 15–17 times higher in cabins compared to homes [8]. 

Despite some progress, many aspects of microbial presence in commercial aircraft have not yet been fully explored. For example, the health-related microbial functional genes, including virulence factors (VFs) and antimicrobial resistance genes (ARGs) have not been characterized. Virulence factors are cellular structures, molecules, and regulatory systems that pathogens use to colonize the host, suppress the host’s immune system, and obtain nutrition from the host [9]. They are a crucial component of pathogenesis. ARGs are genes that protect microorganisms from the effects of antimicrobials. The spread of ARGs poses an increasing threat to public health, causing hundreds of thousands of human death per year globally [10]. Additionally, the health-related metabolites and chemicals are not comprehensively characterized in commercial aircraft. Previous studies used a low-throughput approach to characterize a few targeted microbial metabolites in aircraft cabins, which can only reveal a small proportion of metabolites/chemicals [8]. 

In a previous study, we assessed the impact of various environmental characteristics, including seat surface material, aircraft manufacturer, flight duration, and aircraft size, on the aircraft cabin microbiome, using a PERMANOVA analysis [6]. The results showed that the seat surface material was the strongest environmental factor in shaping microbial diversity (R^2^ = 0.38, PERMANOVA). The other environmental characteristics only had a minor impact on the overall microbial variation (R^2^ < 0.10). Hence, in this study, we primarily focus on the differences in microbial and chemical variations between the two surface types (TSC vs. LSC). The aim of this study is to characterize the abundance of risk microbial genes (VFs and ARGs), microbial metabolites, and synthetic chemicals in commercial aircraft cabins with different surface materials. VFs and ARGs were characterized via shotgun metagenomics, and indoor metabolites and chemicals were characterized via untargeted liquid chromatography–mass spectrometry (LC/MS). The goal is to assess the biological and chemical exposure in commercial aircraft cabins, and provide new insights into disease control and intervention strategies to improve the health of passengers and aircrew. Furthermore, the aircraft exposure will be compared to independent datasets in common school environments to further assess the exposure burden in aircraft. 

## 2. Materials and Methods

### 2.1. Study Design and Dust Sample Collection

Vacuum dust was collected from two airline companies operating at Arlanda airport, Stockholm, Sweden, in 2009. Nine aircraft from each company were selected for the study, with one company using textile seats in cabins (referred to as TSC) and the other using leather seats in cabins (referred to as LSC). The other surface materials were the same in TSC and LSC. The TSC comprised four Airbus 340 flights, three Boeing 737 flights, and two McDonnell Douglas MD-80 flights. In contrast, all nine flights in LSC were Airbus, ranging from models 319 to 330 [6]. The majority of environmental characteristics were the same between TSC and LSC, including cabin temperature (24–27 ℃), relative humidity (10–20%) and air recirculation (35–50%). All aircraft used were stationed at Stockholm’s Arlanda airport and underwent the same cabin cleaning procedure. Approximately two-thirds of the flights were short European flights, and the passengers were predominantly European.

Dust samples were collected when the aircraft were on the ground between flights, and before routine cleaning. The dust was collected using a Siemens Super XS Dino E 1800 W vacuum cleaner, operated at 1200 W, equipped with a special dust collector that had a cellulose acetate Millipore filter (ALK Abello, Copenhagen, Denmark). The filter retained 81% of particles in the range of 0.5–1.0 μm, 95% of particles in the range of 1–10 μm, and 100% of particles larger than 10 μm, effectively capturing the majority of bacteria and other microorganisms. In each aircraft, three dust samples were collected by vacuuming for 4 min in the forward, middle, and after parts of the cabin, including 2 min on the passenger seats and 2 min on the floor. In total, 54 dust samples were collected from aircraft cabins. The dust was sieved through a 0.3 mm mesh screen to obtain the fine dust, which was then immediately stored in a −80℃ freezer until DNA extraction. Finally, the three dust samples collected in each aircraft (forward, middle, and after parts of the cabin) were pooled together.

### 2.2. Shotgun Metagenomic Sequencing

The sequencing and assembly procedure were reported in our previous publication [6], and we provide a brief overview of the procedure in this paragraph. DNA extraction and shotgun metagenomic sequencing were conducted by personal Biotechnology Co., Ltd. (Shanghai, China). Total microbial genomic DNA was extracted from 50 mg fine dust using an E.Z.N.A.Soil DNA Kit D5625-01 (Omega Bio Tek, Inc., Norcross, GA, USA) with bead beating and a spin filter. DNA quality and quantity were evaluated via agarose gel electrophoresis and a NanoDrop ND-1000 spectrophotometer. Negative controls were included to assess contamination. The DNA sequencing library was constructed with Illumina TruSeq Nano DNA LT Library Preparation Kit (Illumina, San Diego, CA, USA) using a paired-end 150 bp shotgun sequencing strategy and an insert size of 400 bp. The library was multiplexed with a dual-indexed barcode structure, and sequenced using the Illumina HiSeq X-ten platform (Illumina, USA). The sequence reads were deposited in the Beijing Institute of Genomics data center (accession number CRA001904) [11]. Raw reads were processed to remove adapters (AGATCGGAAG) by Cutadapt (v1.2.1) [12]. Low-quality reads (<Q20, read accuracy <99%) were trimmed using a 5 bp slide-window algorithm. The trimmed reads with length >50 bp and no ambiguous bases were kept for further analysis. Human reads were removed using KneadData and BMTagger. Sequencing and assembly statistics were reported in our previous publication [6]. Cleaning reads were de novo assembled using MEGAHIT (v1.0.5) with the de Bruijn graph (DBG) algorithm [13]. Genes (>300 bp) in metagenomic scaffolds were predicted using MetaGenoMark (v3.25) [14], and clustered using CD-HIT (v4.8.1) with 90% sequence identity to obtain a non-redundant gene catalog [15]. Gene depth was estimated using SOAPcoverage (v2.7.9) based on the number of aligned reads [16]. 

The remaining analyses were newly conducted in this study. Microbial taxonomy was annotated using MEGAN (v5.0) by searching against the NCBI-NT database with the lowest common ancestor approach (e value < 0.001) [17]. The non-redundant genes were annotated by searching against the KEGG Automatic Annotation Server with the GENES dataset as prokaryotes. The annotated KEGG profiles of the non-abundance genes were then classified into functional categories, and the functional gene categories enriched in TSC and LSC were calculated via linear discriminant analysis effect size (LEfSe, v1.0) [18]. Microbial VFs were annotated using DIAMOND (v2.0.4) against the VFDB database (v2019) [19]. ARGs were annotated using RGI (v5.2.0) against the CARD database (v3.1.3) [20]. Only alignments with pass_bitscore > 600 and best_hit_bitscore > 50 were included. The abundances of the VFs and ARGs were defined as reads per kilobase per million mapped reads (RPKM). LEfSe was used to analyze the enriched VFs and ARGs in TSC and LSC. Mmvec (v1.0.6) was used to estimate microbe–metabolite interaction co-occurrence probabilities [21]. All detected metabolites and the top 50 microbial taxa were included in the analysis.

### 2.3. Dust Metabolites/Chemical Profiling via LC/MS

Untargeted LC/MS analysis was performed to characterize the environmental metabolites and chemicals in 10 mg of vacuum dust by BioNovoGene (Suzhou, China). Chromatographic separation was achieved using an Acquity UPLC HSS T3 Column (2.1 × 150 mm, 1.8 μm, Waters Corporation, Milford, MA, USA) at a temperature of 40℃ with a flow rate of 0.25 mL/min. Detection was carried out using a Q Exactive HF-X Hybrid Quadrupole-Orbitrap Mass Spectrometer (Thermo Fisher Scientific, Waltham, MA, USA). ESI-MSn experiments were conducted with a spray voltage of 3.5 kV in positive mode and −2.5 kV in negative mode. A mass range of m/z 81-1000 was fully scanned using the spectrometer, with a mass resolution of 60,000. An HCD scan was performed for data-dependent acquisition (DDA) MS/MS experiments. A total of 1420 metabolites were characterized using the second stage of mass spectrometry (MS2). The detected analytes were annotated against several public databases, including the Human Metabolome Database (www.hmdb.ca, accessed on 1 July 2021), METLIN (metlin.scripps.edu, accessed on 1 July 2021), mzCloud database (www.mzcloud.org, accessed on 1 July 2021), MoNA (mona.fiehnlab.ucdavis.edu, accessed on 1 July 2021), and MassBank (www.massbank.jp, accessed on 1 July 2021). Differentially enriched metabolites/chemicals were identified using the Mann-Whitney-Wilcoxon test, followed by false discovery rate (FDR) adjustment.

## 3. Results

### 3.1. Microorganisms and Functional Genes in TSC and LSC

The sequencing of shotgun metagenomics generated 446 Gb of raw data and 337 Gb of clean data. After removing human reads, 72.5 Gb of microbial data (2.0–7.4 Gb per sample) was used for further analysis. A total of 6289 microbial taxa were identified, with Proteobacteria (48.2%) and Actinobacteria (30.3%) as the dominant phyla. The top abundant species were *Paracoccus denitrificans* (3.1%), *Micrococcus luteus* (3.0%), unidentified *Sphingomonas* (2.7%), *Variovorax paradoxus* (2.5%), *Propionibacterium acnes* (2.3%), and *Achromobacter xylosoxidans* (2.2%). These microbial species were mainly derived from the outdoor environment based on a comparison against the Earth Microbiome Project [22]. Human-derived species accounted for a smaller proportion, including *Faecalibacterium prausnitzii* (1.3%), *Staphylococcus epidermidis* (1.0%), and *Streptococcus mitis* (0.9%) [23]. Non-metric multidimensional scaling (NMDS) analysis revealed a significant difference in microbial taxonomic composition between TSC and LSC (Figure 1A; Adonis, R^2^ = 0.38, *p* < 0.001). 

A total of 762,865 microbial genes were annotated using the KEGG database. The metabolism pathway accounted for 60.0% of annotated functional genes, followed by pathways in genetic information processing (10.1%), environmental information processing (8.4%), cellular processes (8.2%), human diseases (7.9%) and organismal systems (5.5%). More specifically, the most abundant pathways were carbohydrate metabolism (12.9%), amino acid metabolism (10.8%), and energy metabolism (10.5%; Figure 1B). The high abundant pathways were consistent with the previous study when using EGGNOG as the annotation database [6]. NMDS analysis showed significant differences in functional composition between TSC and LSC (Adonis, R^2^ = 0.21, *p* < 0.001) (Figure 1C). TSC had a higher gene abundance related to replication and repair, nucleotide metabolism, and genetic information processing, while LSC had higher gene abundance related to transcription, cell motility, signaling molecules, and interaction (LDA > 3, *p* < 0.05; Figure 1D). Although there was variation in gene abundance between TSC and LSC, the differences were generally small (e.g., replication and repair gene abundance was 3.9% in TSC and 3.5% in LSC; transcription gene abundance was 0.81% in TSC and 0.97% in LSC). 

### 3.2. Virulence Factors and Antibiotic Resistance Genes in TSC and LSC

We further performed a functional gene analysis by blasting all sequenced reads against the VFDB and CARD databases. The VFs with significantly higher abundance in TSC or LSC were identified using a stringent cutoff (LDA score > 2, *p*-value < 0.01 in LEfSe and abundance fold changes > 1.5 or <0.67). A total of 17 VFs were enriched in TSC, while only one VF was enriched in LSC (Figure 2A; Appendix A). Ten of the enriched VFs in TSC were adherence genes, including non-fimbrial adherence (CBPs), autotransporter adhesion (Hia/Hsf, BadA/Vomp, BoaA and BoaB), surface-associated adhesion (C5a peptidase), cell wall-anchored protein (AS and Lmb), and type IV pili. The abundances of these VFs ranged from 0.2 to 3.0 reads per kilobase per million mapped reads (RPKM). The potential microbial sources of each VF were determined using BLAST on each sequence against the NCBI NT database and assigning the top three results with the highest sequence similarities and bitscores. The majority of the adherence VFs were from facultative pathogens, including *Haemophilus parainfluenzae*, *Veillonella parvula*, *Streptococcus pneumoniae*, *Streptococcus mitis*, and *Streptococcus sanguinis* (Figure 2A; Appendix A). Species abundance analysis indicated that these species were also significantly enriched in TSC compared to LSC (*t*-test, *p* < 0.001), which was consistent with the VF analysis. In addition to adherence genes, TSC was also enriched in VFs related to biofilm and quorum sensing (Fsr and AI-2), exoenzyme (neuraminidase), immune modulation (lipooligosaccharide and IgA1 protease), and nutritional/metabolic factors (Isd and Hpt). The RPKM of these VFs ranged from 0.5 to 3.5. These VFs were primarily from facultative pathogens, including *Klebsiella pneumonia*, *Streptococcus parasanguinis*, *S. pneumonia*, *S. mitis*, *Staphylococcus epidermidis*, and *Staphylococcus aureus*. Only one adherence gene (cytadherence organelle) was enriched in LSC, and this was mainly from *Corynebacterium*.

A total of four ARGs were enriched in LSC (Figure 2B; Appendix A). These ARGs included genes related to antibiotic target alteration (glycopeptide resistance genes, vanSF, and vanRA) and antibiotic efflux (QepA1 and mtrA). The enrichment of these ARGs was driven mainly by two aircraft from LSC (Figure 2B). The enriched ARGs were primarily derived from *Variovorax paradoxus*, *Paracoccus denitrificans*, *Sphingomonas*, and *Massilia*. The RPKM of the enriched ARGs was 0.2 to 1.2, which was lower than the abundance of VFs in aircraft. 

### 3.3. Potential Risk and Protective Chemicals in TSC and LSC

The abundances of metabolites and chemicals was assessed using untargeted mass spectrometry, resulting in the characterization of 1420 metabolites. The results showed that the surface-type materials, TSC and LSC, affected the accumulation of microbial metabolites and chemicals, as evidenced through ordination analysis (Figure 3A,B).

Our epidemiological survey in Malaysia and Shanghai revealed that the presence of indoor indoles, keto acids, and flavonoids was protective against the prevalence of asthma and rhinitis [25]. Further molecular experiments showed that these metabolites could reduce inflammation and oxidative stress [26,27,28]. Therefore, we considered them to be potential protective metabolites, and compared their concentrations in TSC and LSC. Indoleacetaldehyde, a derivative of indole, was found to be enriched in TSC (Figure 3D, Appendix A). Conversely, LSC was enriched with one indole derivative (N-acetylserotonin), three flavonoids (kaempferol, genistein, and 6-hydroxydaidzein), and one keto acid derivative (oxoadipic acid), suggesting that LSC contains a higher concentration of protective metabolites than TSC. Flavonoids are secondary metabolites from various plants, while keto acid and indole derivatives can be produced by microorganisms. To determine the relationship between these metabolites and the top 50 cabin microorganisms, we conducted a neural network co-occurrence probability analysis (Figure 4). The results indicated that several cabin microorganisms, including *Corynebacterium*, *Nocardiodes*, *Streptococcus*, *Sphingomonas*, and *Pseudomonas* co-occurred with oxoadipic acid, indoleacetaldehyde, and N-acetylserotonin, implying that these microorganisms may be responsible for producing these protective metabolites. 

### 3.4. Comparison of Exposure between Aircraft and a Common School Environment

Our findings indicate that TSC had higher levels of hazardous biological (VFs) and chemical exposures (pesticides/herbicides, and detergents) compared to LSC. Two of our studies have assessed indoor microbial exposure in middle school classrooms in China (33 classrooms in 9 schools) and three centers in Malaysia (96 classrooms in 24 schools in 3 locations) using shotgun metagenomics and untargeted metabolomic profiling [25,29]. The indoor dust in classrooms was vacuumed from the floor, tables, and textile curtains. As these studies used the same sequencing protocol and bioinformatics pipeline, we compared the abundances of VFs, ARGs, and chemicals in this study with those in the school environment. 

The VFs showed a trend of TSC > LSC > school environment (Figure 5A). Most VFs in TSC and LSC were significantly higher than in the schools in Shanxi (t-test, *p* < 0.01). Only C5a peptidase, Lmb, and Type IV pili showed similar abundance between aircraft and school classrooms.

For the four enriched ARGs, vanSF, and QepA1 were found to have a higher abundance in the aircraft, compared to schools, while the other two (vanRA and mtrA) had lower abundances in TSC compared to schools (*p* < 0.01; Figure 5B). Overall, the abundance of ARGs was similar between aircraft and schools.

Regarding the levels of pesticide/herbicide chemicals, TSC had significantly higher levels of Benfuracarb, Fenoxycarb, and Propazine compared to schools in Malaysia, while the levels of Carbofuran, Famoxadone, and Bromacil were lower in the aircraft compared to Malaysian schools (*p* < 0.01; Figure 5C). The levels of detergents and cleaning products were significantly higher in the aircraft, with a trend of TSC > LSC > school environment. 

## 4. Discussion

This study has several strengths. Firstly, it is the first to use high-throughput shotgun metagenomics and untargeted metabolomics to examine the presence of VFs, ARGs, metabolites, and chemicals in aircraft cabins. The results showed a higher abundance of VFs and synthetic risk chemicals in TSC compared to LSC (17 vs. 1 for VFs, and 11 vs. 1 for synthetic risk chemicals). Furthermore, previous studies have shown that TSC has 1.8-fold more dust than LSC [8], highlighting a higher exposure burden of VFs and risk chemicals in TSC. Secondly, this study compares the exposure burden in aircraft to common school environments. The abundances of VFs and detergent chemicals is higher in aircraft compared to schools. Both the aircraft and school studies utilized the same DNA extraction, library construction, sequencing protocol, batch normalization, and statistical analysis; therefore, the technical bias was largely controlled. The results indicate that the exposure burden of VFs and detergent chemicals is significantly higher in aircraft compared to the common school environment, which may explain the heightened risk of infectious and chronic inflammatory diseases in aircraft. However, choosing smooth leather surfaces can significantly reduce the exposure burden of VFs and detergent chemicals compared to porous textile surfaces. Thus, selecting appropriate surface materials can serve as a potential remediation and intervention strategy in public transportation to enhance passenger health. 

This study has several limitations that should be considered. Firstly, the metabolites and chemicals were quantified using untargeted metabolomics, which only provides relative abundance and not absolute concentration. As a result, it is not possible to determine if these synthetic chemicals exceed the health regulation limits in aircraft. However, untargeted metabolomics has advantages over other absolute quantification methods such as targeted metabolomics. It is a top-down approach that allows for the characterization of more than 1000 metabolites/chemicals without the need for a prior hypothesis. Targeted metabolomics, on the other hand, focuses on a predefined set of metabolites, and it can only quantify 30 to 40 metabolites/chemicals with high precision and accuracy. As no high-throughput metabolomic survey has been conducted in aircraft, untargeted metabolomics provides a comprehensive overview of the metabolites in this environment. Future studies using targeted metabolomics can provide more accurate information about the absolute concentrations of the metabolites and chemicals characterized in this study. Second, the limitations of using second-generation shotgun metagenomics for characterizing VFs and ARGs should be noted. The short read length may limit the accuracy of the microbial source estimation. While the NCBI NT database was the largest public database available, it is still possible that the true source microorganisms were not recorded in the database, and that only close homologous sequences were identified. As such, the microbial source analysis results should be considered as preliminary exploratory estimations, rather than definitive identifications. Third, the dust samples collected in this study were stored for 8 years before sequencing, which may have resulted in potential degeneration. However, previous studies have reported that most metabolites can be maintained when stored at −80 °C for more than 7 years [30], suggesting that the results of this study should be reliable. Lastly, the sample size of this study was relatively small, with only 18 aircraft characterized. Three dust samples were collected from the forward, middle, and after parts of the cabin in each aircraft, and they were pooled together for high-throughput metagenomics and metabolomics sequencing. This means that the collected dust represents the exposure in the entire aircraft cabin, and not just a specific location. To support the findings, stringent statistical thresholds were set in the statistical analysis using the LEfSe and non-parametric test (LDA > 3, *p*-values < 0.01), indicating that the results of this study should be reliable. 

This study found that TSC had a higher abundance of virulence factors than LSC and the school environment, specifically VFs related to adherence, biofilm formation, and immune modulation. The functional mechanisms of the reported VFs have been studied in epithelial cell lines and animal models. For example, type IV pili genes are crucial for the initial attachment, motility, and competence of microorganisms, while choline-binding proteins (CBPs) play a role in the distribution and restructuring of choline in the cell wall, affecting the pathogen–host interaction [31]. Laminin-binding protein (Lmb) is an extracellular protein that mediates attachment to human laminin, and Hia/Hsf has high-affinity adhesive activity and interacts with a range of respiratory epithelial cells [32]. However, the impacts of different materials on the attachment and adherence properties of VFs remain largely unknown. A study found that methicillin-resistant *S. aureus* (MRSA) decreased more quickly on smooth surfaces such as plastic tables and metal toilet handles than on the porous textile seats in aircraft cabins [33]. This result supports our findings, but the study only characterized MRSA, not VFs, and it is unclear which VF causes the variation in MRSA attachment. Textile materials have more surface area and porous structures, making them favorable sites for microorganism colonization and chemical deposition. We also observed higher levels of opportunistic pathogens, including *S. mitis*, *S. pneumoniae*, *Streptococcus pseudopneumoniae*, *Staphylococcus epidermidis*, *S. aureus*, and *Haemophilus parainfluenzae*. The interconnections between these opportunistic pathogens, adherence VFs, and textile surfaces suggest that VFs may play a key role in microbial abundance variation in aircraft cabins. Therefore, it is important to investigate the mechanisms and functions of VFs in the transmission and persistence of pathogens on commonly used manufactured materials. This information can inform better choices of surface materials for indoor environments and improve passenger health in public transport systems. 

Only four ARGs were enriched in LSC, primarily from the genera *Variovorax*, *Massilia* and *Sphingomonas*. The abundance of the enriched ARGs (0.2–1.2 RPKM) was lower compared to the VFs (0.5–3.5 RPKM). Additionally, the abundance of ARGs in aircraft was found to be similar to the levels present in the school environment, suggesting that the spread of ARGs is not a major concern in aircraft cabins. ARGs are widely spread in various indoor environments [34], but their concentration is generally low, except in hospitals and intensive care units where the intensive use of antibiotics can increase the spread and concentration of ARGs [35,36]. However, commercial aircraft cabins do not extensively use antibiotics, and thus, the low levels of ARGs are in line with expectations. 

In this study, five detergents showed an abundance trend of TSC > LSC > school environment, suggesting a high exposure burden of detergents in aircraft with textile surfaces. These detergent chemicals have been shown to have adverse health effects in humans, as stated in GHS statements, the PubChem database, and laboratory experiments. For instance, triethanolamine, diethanolamine, and diethyl phthalate have been linked to skin irritation and allergic dermatitis through epidemiological and animal studies [37,38,39]. An inhalation toxicity study in rats showed that triethanolamine can cause changes in the laryngeal epithelium and inflammation [40]. The high concentration of detergents in aircraft may result from frequent cleaning practices, and the porous nature of the textile surfaces may retain more detergents than smooth leather surfaces. This deposition of risk chemicals in textiles can release into the air and increase the likelihood of developing chronic inflammatory diseases, and skin or airway irritation in passengers. Thus, it is essential to regulate or to be cautious with the use of detergents and cleaning products in public transport systems, particularly in cabins with porous textile surfaces.

In addition to potential risk genes and chemicals, this study also profiled protective metabolites in aircraft, including indoles, keto acids, and flavonoids. The findings showed a higher abundance of protective metabolites in LSC (5) compared to TSC (1). These metabolites were defined based on previous epidemiological, microbiome, and metabolic/chemical surveys, and molecular experiments support their health benefits. For instance, N-acetylserotonin can reduce oxidative stress and inflammation by inhibiting the expression of NF-κB [41]. Kaempferol, genistein, and hydroxydaidzein can also mitigate chronic inflammation by lowering inflammasome activation and reducing the expression of inflammatory cytokines, including NF-κB, TNF-α, and IL-6 [42,43,44]. These results suggest that leather surfaces might be preferable to textile surfaces in commercial aircraft. 

## 5. Conclusions

In this study, we conducted the first comprehensive, high-throughput analysis of the multi-omic microbiome and metabolome in commercial aircraft cabins. Our findings show that the exposure burden to microbial virulence factors and detergents is significantly higher in aircraft compared to the typical school environment, increasing the likelihood of infectious and chronic inflammatory diseases in these settings. However, the results suggest that the use of smooth leather surfaces can significantly reduce exposure to these harmful factors, compared to porous textile surfaces. Thus, choosing the right surface materials could be an effective strategy for improving the health of passengers in public transportation.

## Figures and Tables

**Figure 1 metabolites-13-00343-f001:**
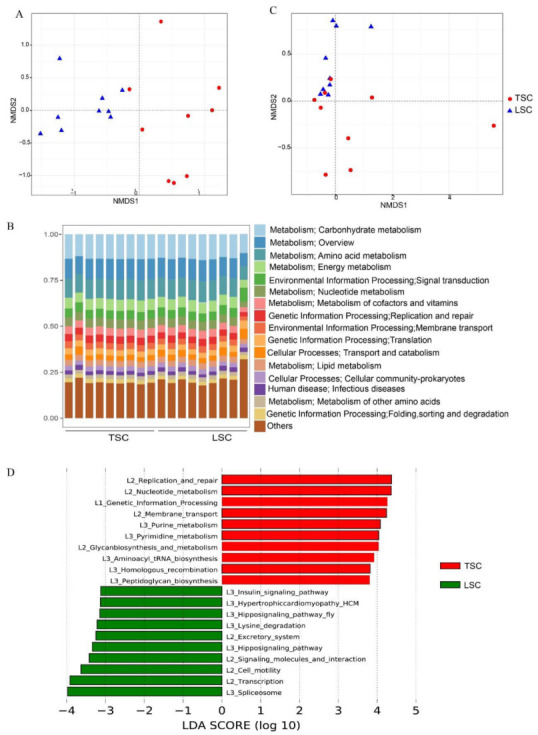
Composition and variation of microbial taxa and functional genes in dust of TSC and LSC aircraft. (**A**) NMDS ordination plot of aircraft microbial taxonomic composition, based on the Bray-Curtis distance matrix. (**B**) Abundance of functional pathways at second-level classification, according to the KEGG database. (**C**) NMDS ordination plot of aircraft functional gene composition, based on the Bray-Curtis distance matrix. (**D**) Characteristic pathways enriched in TSC and LSC aircraft, as determined with LEfSe analysis (LDA > 3 and *p*-value < 0.01).

**Figure 2 metabolites-13-00343-f002:**
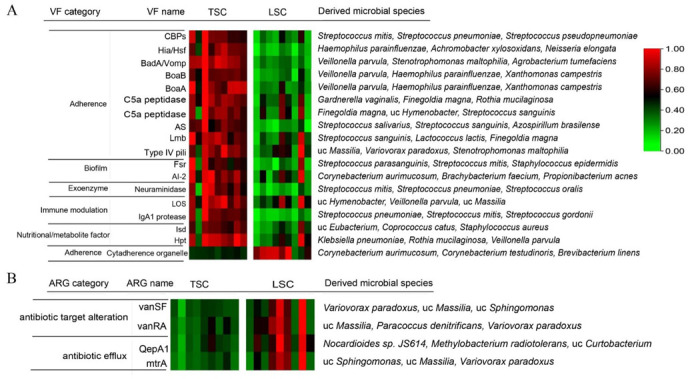
Relative abundances of virulent factors (VFs) and antibiotic resistance genes (ARGs) enriched in TSC and LSC. Only significantly enriched (**A**) VFs and (**B**) ARGs in TSC and LSC are shown (LDA > 2, *p*-value < 0.01 in LEfSe analysis, and abundance fold changes >1.5 or <0.67). Heatmaps show the relative abundances of VFs and ARGs, with the most abundant being defined as 1. The potential microbial sources of each VF or ARG were identified by searching all sequences against the NCBI-NT databases. The three most abundant microbial sources are listed in the table as the main sources of the VFs/ARGs.

**Figure 3 metabolites-13-00343-f003:**
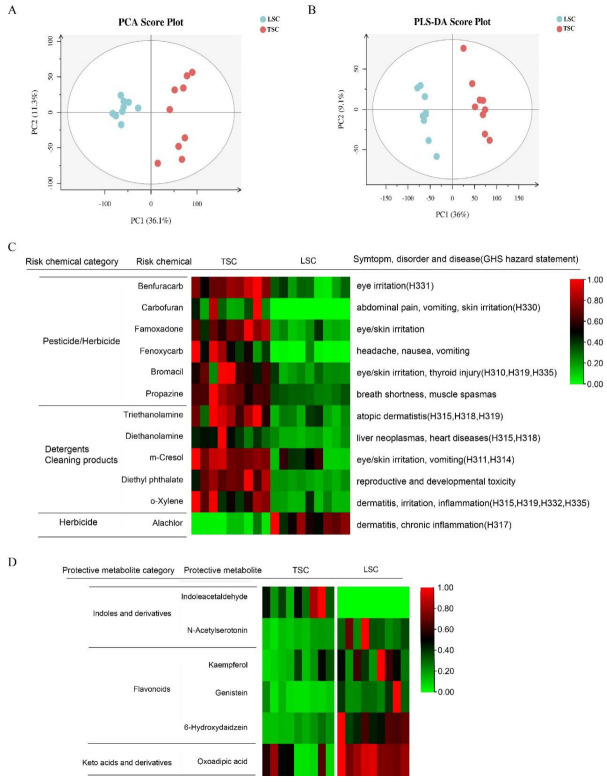
Variation of metabolites/chemicals in TSC and LSC. (**A**) Principal Component Analysis (PCA) and (**B**) Partial Least Squares Discriminant Analysis (PLS-DA) plots of aircraft metabolites/chemicals. (**C**) Risk chemicals and (**D**) protective metabolites enriched in TSC and LSC (*p* < 0.01, FDR < 0.1 and fold change >2). The relative abundance is displayed, with the highest value set as 1 in each line, and the remaining values calculated as ratios. The potential risk chemicals were defined as those associated with symptoms, disorders, and diseases according to Pubchem, or having GHS hazard codes between H300 and H336 (airway, skin, and eye irritation). These chemicals were mostly synthetic. Protective metabolites, including flavonoids, isoflavonoids, indoles, and keto acids, were identified based on previous studies. Chemicals annotated as drugs were excluded from the table. H311: Toxic in contact with skin; H314: Causes severe skin burns and eye damage; H315: Causes skin irritation; H317: May cause an allergic skin reaction; H318: Causes serious eye damage; H319: Causes serious eye irritation; H330: Fatal if inhaled; H331: Toxic if inhaled; H332: Harmful if inhaled; H335: May cause respiratory irritation. Potential risk chemicals in TSC and LSC were defined as those with GHS hazard codes (H300 to H336) or having adverse symptoms, disorders, and diseases, as found in Pubchem and Pubmed. Eleven potential risk chemicals were identified in TSC, while only one was found in LSC (Figure 3C; Appendix A). The risk chemicals in TSC included six pesticides/herbicides (benfuracarb, carbofuran, fenoxycarb, bromacil, propazine, and famoxadone) and five detergent and cleaning product chemicals (triethanolamine, diethanolamine, m-cresol, diethyl phthalate and o-xylene). These chemicals can cause symptoms such as eye and skin irritation, vomiting, headache, inflammation, and dermatitis. Furthermore, m-cresol, triethanolamine, and o-xylene are considered potential human carcinogens according to assessments from the Integrated Risk Information System (IRIS) and the Carcinogenic Potency Database (CPDB) [24]. In LSC, only one potential risk chemical, alachlor (an herbicide), was identified, with adverse health effects including chronic inflammation and skin irritation.

**Figure 4 metabolites-13-00343-f004:**
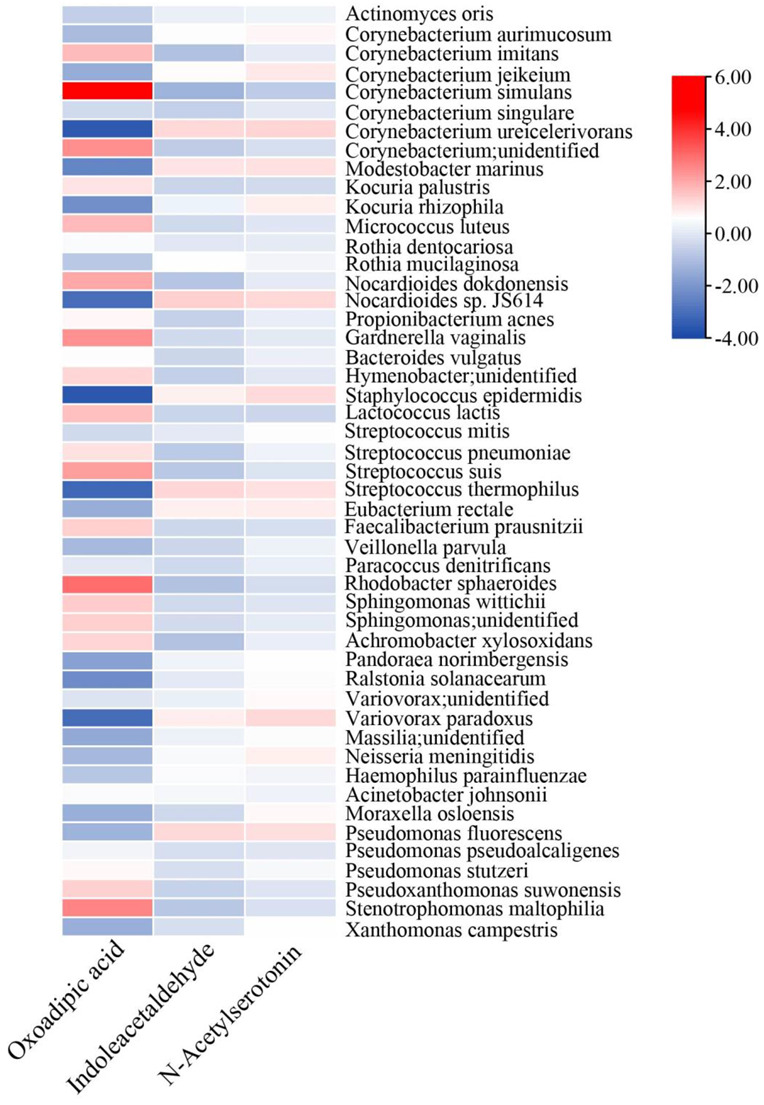
Co-occurrence probability of aircraft microbial taxa and protective metabolites in TSC and LSC. The top 50 microbial taxa present in aircraft cabins are displayed on the *y*-axis of the heatmap, and the potential protective metabolites (indole and keto acids) are displayed on the *x*-axis. The co-occurrence probabilities between the microorganisms and metabolites were estimated using mmvec. A positive value indicates a strong likelihood of co-occurrence between the microbe and metabolite.

**Figure 5 metabolites-13-00343-f005:**
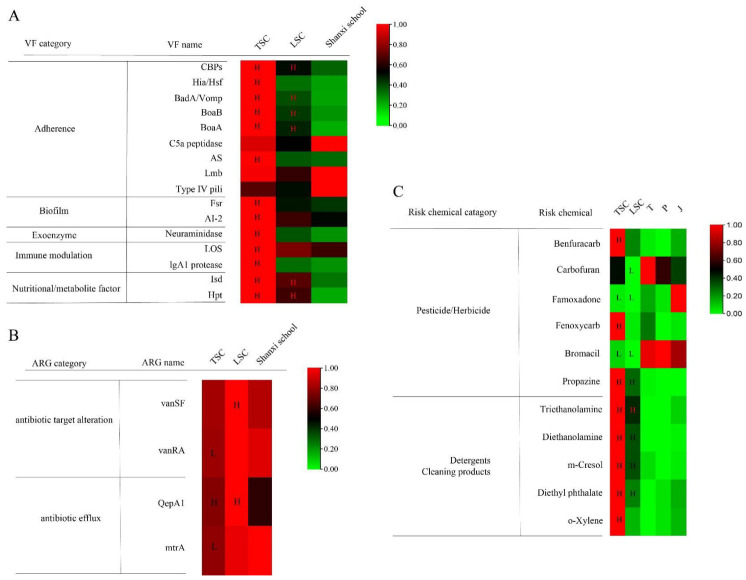
Comparison of relative abundances of (**A**) virulent factors (VFs), (**B**) antibiotic resistance genes (ARGs), and (**C**) synthetic chemicals in aircraft cabins and school classrooms. The most abundant VF, ARG, and chemical in each line is defined as 1. The abundances of VFs, ARGs, and chemicals in TSC and LSC were compared with the schools in Shanxi, China (N = 33); and Terengganu (T), Penang (P), and Johor Bahru (J), Malaysia (N = 96). Significantly higher abundance in TSC/LSC compared to schools is indicated by an “H” symbol in the heatmaps, while significantly lower abundance in TSC/LSC is indicated by an “L” symbol (t-test, *p* < 0.01).

## Data Availability

The sequence reads were submitted to the Genome Sequence Archive in the Beijing Institute of Genomics (BGI, http://english.big.cas.cn/) data center, with accession number CRA001904.

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
