# Peer review of "Microbial Virulence Factors, Antimicrobial Resistance Genes, Metabolites, and Synthetic Chemicals in Cabins of Commercial Aircraft"

_metabolites, 2023, doi:10.3390/metabo13030343_

Round 1

Reviewer 1 Report

The study is interesting and important because it allows better choice of surface materials for indoor environments and as a result to improve passenger health in public transport systems. The authors studied the spread of virulence factors, antimicrobial resistance genes, as well as metabolites and synthetic chemicals in the cabins of aircraft with textile and leather surfaces. A wide range of bioinformatics analysis tools and untargeted mass spectrometry were used, which allowed for a comprehensive analysis.

Nevertheless, the manuscript needs a revision before publication. Please consider my suggestions as listed below.

Comments

Please follow exactly the instructions for Authors when preparing a manuscript.

In the text, reference numbers should be placed in square brackets [ ], for example [1], [1–3] or [1,3].

The “Author Contributions” and “References” parts must be written according to the requirements of the journal.

The abbreviation should be used after the full term. Please be consistent with the usage of all abbreviations.

Subsection 3.1. Environmental characteristics in TSC and LSC

I recommend that the first paragraph be moved to Methods as Environmental characteristics in TSC and LSC.

The second paragraph is more for the Introduction section

Minor comments

Line 192-193 italics for Latin names of bacteria species

Figure 1. The designations for parts 1B and 1C are mixed up

Line 242 italics for Latin names of bacteria species

Reviewer 2 Report

The manuscript describes the presence of virulence factors, antimicrobial resistance genes, metabolites and synthetic chemical in cabins of commercial aircraft and compares with schools, which is really relevant due to a few studies with this scope have been done and generated information is really relevant to take preventive measures and avoid spreading of pathogens and infections.

One of the main concerns is that in the manuscript is described the methodology to take dust samples, extract DNA, sequencing and until de novo assembly are the same than their previous work Sun et al. 2020. As it was drafted the manuscript, authors mention citation but it is seemed that perform all the methodology again when the novel work starts with the different pipeline to analyze assembled genes. I considered it is important to clarify this point to be well stated.

In material and methods no mention what environmental characteristics were measured but they present in results.

184 the values of raw reads and clean data differs from previously published, although identified the 7437 taxa previously reported.

205-218 these analyses were done previously at Sun 2020 but due to different data treatment results differs from previous, however, there is no comparison to contrast if any differences exist.

Some specific corrections

Line 77 citation is lacking

105 add space 0.3mm

108 add period

242 Corynebacterium in italics

In figure 4, indoleacetaldehyde and N-acetylserotonine are lighter than oxoadipic acid column, please correct.
